# Changes in Subjective Time and Self during Meditation

**DOI:** 10.3390/biology11081116

**Published:** 2022-07-26

**Authors:** Damisela Linares Gutiérrez, Stefan Schmidt, Karin Meissner, Marc Wittmann

**Affiliations:** 1Institute of Frontier Areas of Psychology and Mental Health, 79098 Freiburg, Germany; damisela.linares.gutierrez@uniklinik-freiburg.de (D.L.G.); stefan.schmidt@uniklinik-freiburg.de (S.S.); 2Palliative Care Unit, Department of Internal Medicine, Medical Faculty, University of Freiburg, 79106 Freiburg, Germany; 3Department of Psychosomatic Medicine and Psychotherapy, Medical Center-University of Freiburg, Medical Faculty, University of Freiburg, 79104 Freiburg, Germany; 4Division of Integrative Health Promotion, Department of Social Work and Health, Coburg University of Applied Sciences, 96450 Coburg, Germany; karin.meissner@hs-coburg.de

**Keywords:** time perception, sense of self, present-moment awareness, meditation

## Abstract

**Simple Summary:**

Meditation induces an altered state of consciousness, which is often described by meditators as being in the present moment and losing one’s sense of time and self. Few studies have assessed these experiences. We invited 22 experienced meditators to participate in two experimental sessions lasting 20 min each (1) to meditate and (2) to read a story as a control condition. We measured their heart and breathing rates during these two sessions and conducted a metronome task before and after each session. In this task, participants had to group metronome beats into perceptual units, a measure of the duration of the present moment. In comparison to the reading condition, the heart and breathing rates showed a mix of increased as well as decreased bodily activity in the meditation condition. In the meditation condition, participants subjectively perceived their body boundaries less strongly, paid less attention to time, and felt time pass more quickly compared to the control condition. No differences between conditions were apparent for the metronome task. This study is the first to show how the sense of self and time are relatively diminished during meditation.

**Abstract:**

This study examined the effects of meditative states in experienced meditators on present-moment awareness, subjective time, and self-awareness while assessing meditation-induced changes in heart-rate variability and breathing rate. A sample of 22 experienced meditators who practiced meditation techniques stressing awareness of the present moment (average 20 years of practice) filled out subjective scales pertaining to sense of time and the bodily self and accomplished a metronome task as an operationalization of present-moment awareness before and after a 20 min meditation session (experimental condition) and a 20 min reading session (control condition) according to a within-subject design. A mixed pattern of increased sympathetic and parasympathetic activity was found during meditation regarding heart-rate measures. Breathing intervals were prolonged during meditation. Participants perceived their body boundaries as less salient during meditation than while reading the story; they also felt time passed more quickly and they paid less attention to time during meditation. No significant differences between conditions became apparent for the metronome task. This is probably the first quantitative study to show how the experience of time during a meditation session is altered together with the sense of the bodily self.

## 1. Introduction

In altered states of consciousness, the three state dimensions, attentional awareness, affect, and time, can reach extreme levels leading to distortions of the self [1]. A joint dissolution of the feelings of self and time is experienced during peak states of meditation, culminating in the feeling of ‘selflessness’ and ‘timelessness’ [2,3,4,5]. Quantitative studies on the experience of time pertaining to a complete meditation session in very experienced meditators have not previously been systematically conducted. In one study with individuals not specifically experienced in meditation, 13 min and 20 s of yogic mindfulness meditation (Shavasana) led to a relative underestimation of duration and a faster subjective passage of time in comparison to listening to classical music for the same duration [6]. A second study with novice meditators corroborated the impression that the duration of meditation feels comparatively shorter when attention is continuously focused on an object of meditation as compared to when meditation is induced by monitoring self-related perceptions and emotions [7]. In a different set of studies with a specific type of meditation, Depth Relaxation Music Therapy (DRMT), participants repeatedly estimated silence after the DMRT condition to have lasted longer than after a control condition [8,9].

A few studies have investigated how time perception in the seconds-to-minutes range is altered before and after meditation. One such study employed a pre- and post- meditation design and compared meditation to a control task undertaken with a duration-production task in the seconds range. This finding indicated specific differences between meditation techniques in timing performance [10]. In another study, participants without prior experience in meditation but who were exposed to a pre-recorded meditation session of 10 min were tested using the classical temporal-bisection task in a time range of around 1 s [11]. The meditation group relatively overestimated the duration of stimuli post- vs. pre-meditation compared to the control group, who did not show a time-perception difference before vs. after listening to a short story for 10 min. A similar effect showing the expansion of subjective duration in experienced mindfulness meditators was seen with a duration-estimation task using stimuli lasting between 4 and 8 s [12]. We were interested in how experienced meditators would relate their impression of time to their sense of self during meditation sessions. Experienced meditators typically report that the sense of time together with the sense of self is diminished, in that the meditation session overall passes subjectively very quickly. During the first minutes of meditation induction, the focused attention on the bodily self might lead to a slow passage of time, but the whole session is typically judged to have passed relatively quickly for experienced meditators, as the sense of self is diminished.

In mindfulness-type meditation practices, meditators continuously try to bring awareness to each present moment with an accepting and non-judgmental attitude [13]. Segmental processing mechanisms are assumed to create temporal ‟windows” of experienced “nowness”, which provide the logistical basis for the experience of the present moment [14,15,16]. The question is how to operationalize assumed altered present-moment awareness during meditation. Linares Gutiérrez et al. [17] recently assessed the effects of meditative states on present-moment awareness in beginners and moderately experienced mindfulness meditators. We found evidence for meditation-induced changes related to the general notion of the embodiment of mental functioning. At the beginning of a meditation session, meditators feel a subjective slowing of the passage of time and an expansion of the present moment, which is achieved through a transiently increased attentional focus on body states leading meditators to be more strongly aware of their bodily selves [18,19]. The representation of time and self is reduced later during the meditation experience [2,3,4]. We employed the metronome task where participants automatically integrate metronome beats into a structural whole (clusters of 1-2, 1-2, or 1-2-3, 1-2-3…) to assess the duration of the experience of the present moment [20]. Multiple path analyses found that meditation led to an increased duration of integration intervals at the slowest metronome frequency (inter-stimulus interval, ISI = 3 s). Physiologically speaking, higher heart-rate variability was related to longer integration intervals at the fastest frequency (ISI = 0.33 s), and the higher the breathing rate during meditation, the larger the integration of intervals at ISI = 1 s. These findings are the first quantitative indications for meditation-induced changes in the subjectively experienced present moment.

The purpose of the present study was to assess the influences of meditative states and related autonomic physiological changes (heart-rate variability and breathing rate) on the present-moment experience, subjective time, and awareness of the physical self in meditators with even more meditation experience than in the previous study [17]. We chose more experienced meditators because we expected to find stronger effects for changes in the sense of self and time as well as for the subjectively experienced present moment. Losing sense of body boundaries and the sense of time is often reported by very experienced meditators. Previous findings on physiological correlates during meditation with experienced meditators indicate both an increase and a decrease in parasympathetic and sympathetic parameters [21,22,23]. Despite a subjectively felt relaxation response in many relaxation meditation techniques, active cardiac responses (i.e., increases in mean heart rate) were recorded and referred to as the ‟meditation paradox” [24]. Therefore, we investigated experienced meditators practicing meditation techniques focusing predominately on the awareness of the present moment. In two separate experimental sessions, participants were asked to either meditate for 20 min in our laboratory performing their own meditation technique or to read a story for 20 min (within-subject design). The performance on a metronome task was compared before and after the two different conditions (a 20 min meditation session vs. 20 min spent reading the story). Meditation depth, self-reported perception of one’s own body (intensity and salience of body boundaries), and experienced time were assessed during both conditions. We hypothesized that mindfulness–meditation states (1) are related to changes in autonomic activity (several indices of heart-rate variability and complexity, e.g., breathing rate), (2) yield alterations in the duration of the present moment as assessed with the metronome task (a widening of heard metronome intervals), and (3) change subjective time and perceived salience of body boundaries (i.e., they become less pronounced).

## 2. Materials and Methods

### 2.1. Design

The study involved a within-subject design including two assessment time points. Participants underwent either a 20 min meditation session or a relaxed 20 min reading session (counterbalanced order across participants) on two consecutive days. Physiological measures (heart rate and breathing rate) were recorded during each session. Performance in the metronome task assessing present-moment awareness was compared before and after each condition (four measurements). Subjective time and the bodily self were assessed with questionnaires following each session.

### 2.2. Participants

Descriptive statistics for the study group are summarized in Table 1. The study included 22 long-term meditation practitioners (12 females, 10 males; mean age: 47.2 years; SD = 12.5 years; age range: 26–67 years). Because this study was the first to use our measures with extremely experienced meditators, we did not have sufficient a priori knowledge for computing the sample size for this specific participant group. Initially, we intended to include n = 32 meditators. This number of participants is required assuming one-tailed calculations for hypothesized differences with an effect size of 0.6, having a standard power of 0.8 with an α-error probability of 0.05. During the study, strong effects were assumed, as in the meditation setting with meditation-naïve participants the subjective scores of time passage reached a Cohen’s d of 0.624 [9]. Due to the COVID-19 pandemic and the strict nationwide and consequent institutional regulations, we were only able to test n = 22 participants during the funding period. Participants were overall well educated (68% with a university degree). Only meditators practicing meditation techniques predominately focusing on awareness of the present moment were recruited (secularized forms of mindfulness meditation: 2; Vipassana meditation: 8; Zen: 8; Tibetan Buddhism: 4). Inclusion criteria included engaging in regular meditation practice for at least 5 years with at least two meditation sessions per week occurring during the previous two months (years of meditation experience: mean = 19.2; SD = 14.83; meditation sessions per week in the last two months: mean = 4.0; SD = 2.08). The amount of formal meditation training was calculated by adding the number of hours of daily sitting-meditation practice and the number of hours of sitting meditation spent in meditative retreats accumulating in lifetime hours of formal meditation training (mean = 2994.2; SD = 3213.4). Participants were recruited by advertisements in meditation centers and by word of mouth; they were fluent in German and reported no history of neurologic or psychiatric disorders. Individuals received financial compensation of EUR 25 for taking part in the two study sessions lasting roughly one hour each. The study was approved by the local Ethics Committee of the Institute for Frontier Areas of Psychology and Mental Health (IGPP, Freiburg, Germany; IGPP_2018-01). All participants provided written informed consent prior to data collection.

### 2.3. General Procedure

The study was conducted in a quiet laboratory room at the Institute for Frontier Areas of Psychology and Mental Health in Freiburg, Germany. Each participant was tested individually. On the first testing day, participants provided written informed consent and filled in one questionnaire regarding socio-demographic variables and one detailing meditation practice. Heart and breathing rates were recorded using a mobile three-channel recording device during both sessions (see below). After having spent about 10 min filling out questionnaires, participants underwent a 5 min baseline procedure in which they were asked to sit in the same position as during the meditation session but not to meditate (physiologic baseline recording). Participants were requested to stay calm during these 5 mins. The metronome task was then presented (*t*1), followed by the 20 min experimental condition (meditation) or the control condition (reading session) in a counterbalanced order. The metronome task was then repeated after each condition (*t*2). At the end of each session, participants filled in scales referring to bodily states and the passage of time experienced during meditation and the reading session. A meditation-depth questionnaire was only handed to the participants after the meditation session.

### 2.4. Conditions

In the meditation condition, participants were asked to meditate for 20 min using their own meditation techniques. All participants practiced sitting meditation using a cushion or a bench placed on a meditation mat. A timer application (Bodhi Timer) playing a singing-bowl sound was used to indicate the beginning and the end of the meditation session.

In the reading condition (control condition), participants were asked to read an excerpt of the German translation of the Dutch novel *The Detour* by Gerbrand Bakker using their own tempo for 20 min while adopting the same position as in the meditation session. The excerpt was presented on a laptop display in the form of a Microsoft Word document. Participants were able to slide through the text using their own tempo and without moving. We chose the excerpt of Bakker’s novel, which contains neutral content and third-person narratives, to keep individuals focused on the task and avoid emotion-related physiologic reactions [17]. Participants remained alone in the room during both conditions.

### 2.5. Physiological Recording

An electrocardiogram (ECG) was recorded with the eMotion FAROS 360° mobile device (Mega Electronics Ltd., Kuopio, Finland), a three-channel ECG recorder which also records breathing activity with an integrated 3D accelerometer placed on the chest. The ECG data were acquired at a 500-Hz sampling rate using two disposable Ag-AgCl electrodes positioned according to a modified Lead II Einthoven configuration: one was placed centrally under the right clavicle, and the other one on the 11th left intercostal space. The electrodes were connected to the eMotion FAROS 360° mobile device. Breathing data were acquired at a 400-Hz sampling rate and at the dynamic range +/−4 g (1 g = 9.81 m/s^2^). The accelerometer was positioned at the level of the 3rd intercostal space on the right side of the sternum.

### 2.6. Questionnaires and Analogue Scales

#### 2.6.1. Meditation Depth Questionnaire (MEDEQ)

The Meditation Depth Questionnaire (MEDEQ, [25]) is a 30-item self-report questionnaire assessing an experienced practitioner’s depth of meditation. Answers are rated on a five-point Likert-type scale ranging from “not at all” to “very strong”. This questionnaire covers five different aspects of meditation experiences: hindrances (e.g., restlessness), relaxation (e.g., calmness), concentration (e.g., attentive control over the mind), transpersonal qualities (e.g., feeling connected, bliss, and grace), and non-dual qualities (e.g., subject/object transcendence). The global score was used as an index of meditation depth.

#### 2.6.2. Perceived Body Boundaries Scale

The Perceived Body Boundaries Scale (PBBS; [26]) is a visual analogue scale (VAS) assessing the salience of the boundaries between the self and the world using a seven-point Likert scale (0 = weak boundary, 6 = strong boundary). Participants indicate the extent of the boundary between themselves and the world. A high score on this scale indicates a strong feeling of separation between one’s self and the world. A low score signifies a weak distinction between the self and the world.

#### 2.6.3. Inventory on Subjective Time, Self, and Space (STSS)

The STSS is an instrument comprising three visual analogue scales assessing the intensity of the subjective perception of one’s own body and experienced time. This instrument was developed in our laboratory and has been included in several studies, including assessments of alterations through meditation [8,17]. The body scale refers to the perception of one’s own body and consists of a seven-pictogram rating scale illustrating a manikin ranging from a highly visible figure to an almost invisible one. Participants were asked to evaluate how intensively they had perceived their bodies during both conditions by choosing one of the seven possibilities (values: 0–6). A high score represents a stronger awareness of one’s own body. The experienced-time scale comprises two different questions: “How intensively did you perceive time during the session?” and “How fast did time pass for you during the session?” These questions are measured with a 100 mm VAS where the minimum and maximum anchor points represent “not intensively at all”, “very intensively”, “very slowly”, and “very quickly” for the two questions, respectively.

### 2.7. Metronome Task

The metronome task was made up of unaccented isochronous auditory clicks (beats). Each beat lasted 19 milliseconds. Sequences of beats comprised the following inter-stimulus intervals (ISI): 2, 1.33, 1, 0.5, and 0.33 s and lasted 15 s. Participants were asked to let an accentuated rhythmic pattern emerge spontaneously (e.g., 1-2, 1-2, or 1-2-3, 1-2-3, etc.) and to report the number of beats the rhythmic pattern contained using a computer keyboard. They were also asked to respond as soon as the rhythmic pattern emerged. The temporal extent of the rhythmic pattern was calculated from the number of integrated beats into one rhythmic group. The dependent score of `temporal integration’ (TI) was defined as the number of beats reported by the subject multiplied by the ISI [20]. Thus, a group period is defined as the period between group onsets (with a presented ISI of 2 s and a reported grouping of two, a group period of 4 s is assumed) [27]. The electroencephalogram (EEG) showed that a sustained brain response is initiated at each of the accents (the respective group onsets) when participants listen to isochronous metronome sounds, and an accent is heard on every second metronome beat [28].

The experiment consisted of a training phase and a test phase. On the first day, participants underwent the training phase before accomplishing the test phase at t1. The training consisted of three stimulus frequencies (1.33, 1, and 2 s ISI) presented twice in random order, resulting in six trials. The test phase was composed of five stimulus frequencies (2, 1.33, 1, 0.5, and 0.33 s ISI), each presented five times in random order, resulting in 25 trials. The central tendency measure for a given frequency corresponding to each of the five trials was calculated as the median of responses per frequency.

### 2.8. Data Analysis

The data for all physiological and psychological variables, the basis for the statistical calculations in the Results section, can be found in the Appendix A.

#### 2.8.1. Breathing Rate

We extracted the respiration periods from the breathing signal during the two 20 min sessions using the peak-detection function available in AcqKnowledge 3.7.2. These signals were visually examined, and artefacts were manually corrected. Respiratory periods were resampled at 15.625 samples per second, employing a linear interpolation method. A band-pass filter using a Hanning window and a low- and high-frequency cutoff of 0.08 Hz and 0.42 Hz, respectively, was applied. Breathing rates (BR) as mean period intervals (in seconds) were computed and averaged over each of the 20 min recording sessions.

#### 2.8.2. Heart Rate

The ECG signals and time intervals between successive R-peaks (RR intervals) were extracted using Kubios HRV Analysis Software 2.0 (Kuopio, Finland) across the two 20 min sessions. Data were first visually examined to control for ectopic and missing beats and automatically corrected by choosing the appropriate threshold level for artefact correction accessible in Kubios. The smoothness-of-priors method was subsequently used to detrend the signal [29]. Following data extraction, we calculated (1) the root mean square of successive differences (RMSSD) in the time domain and (2) the high-frequency (HF) spectrum in the frequency domain. We also computed the detrended fluctuation analysis (DFA), which yields two different coefficients: (3) alpha 1 (α 1, short-term scaling), and (4) alpha 2 (α 2, long-term scaling). We finally applied the nonlinear method for assessing regularity and complexity changes of the heart-rate time series, (5) the approximate entropy (ApEn), and (6) the sample entropy (SampEn). These indices are explained in the following paragraph.

We computed the root mean square of successive differences (RMSSD) in the time domain and the high-frequency (HF) spectrum in the frequency domain to assess the heart-rate variability. According to the recommendations for heart-rate variability (HRV) assessment by Laborde et al. [30], the parameters (RMSSD and HF) are best indicated when the focus of interest lies on vagal modulation. Time-domain (RMSSD) and frequency-domain (HF component) measurements of the HRV have been shown to correlate [31,32]. Evidence exists that these indices might not convey the same biological meaning [33]. Therefore, we proceeded to examine vagal changes in the HRV using both indices. The RMSSD in the heart rate is a time-domain HRV parameter indicating short-term variability in the heart rate (HR). It is the most used time-domain measure applied to estimate parasympathetic activity [34,35], and it is considered to be free of respiratory influences [36,37]. In the frequency domain, a Fast Fourier Transformation (FFT) is calculated to measure the relative power of the HF spectrum (HF[%]: HF[m^2^]/total power[m^2^] × 100%) of the RR intervals series. The HF band between 0.15 and 0.40 Hz is considered to reflect vagal activity [38] and, in contrast to the RMSSD, it is thought to be influenced by breathing when the cycles per minute are below 9 or above 24 [38,39]. To guarantee the reliability of the HF components, we used the breathing-interval average (BR) to control for respiratory rates below 9 or above 24 cycles per minute, which refers to the 0.15- and 0.40-Hz band regarding HFs. Participants whose breathing rate exceeded these limits were excluded from the frequency domain HRV analysis.

A further method to assess the HRV is to utilize nonlinear parameters, which represent complex interactions between the autonomic nervous system (ANS) and the central nervous system (CNS) [38]. A commonly used nonlinear method is detrended fluctuation analysis (DFA), which uses two different coefficients, alpha 1 (α 1, short-term scaling) and alpha 2 (α 2, long-term scaling), to quantify the degree of fractal correlation properties of heart-rate time series and it is referred to as consecutive inter-beat intervals [24]. While α 1 relates to sympathetic activity, α 2 reflects sympathetic and parasympathetic activities.

An additional method to analyze heart-rate series, approximate entropy (ApEn), is a nonlinear method that measures the regularity and complexity changes of the heart-rate time series. ApEn is related to parasympathetic modulation. Lower levels of ApEn result from close successive time series and indicate high regularity and little complexity [40]. Sample entropy (SampEn) was developed later as an improved version of ApEn [41].

## 3. Results

### 3.1. Between-Condition Differences in Physiological Variables

We applied a double-analysis approach to the physiological data. First, we ran repeated-measures, 2 × 2 ANOVAs with the factors of condition, the meditation session vs. the reading session, and the factor time interval (5 min baseline vs. 20 min session). The means and standard deviations for the four cells referring to the 2 × 2 ANOVA for each variable (BL, experimental session; meditation, reading story) can be read in Table 2 and Table 3. The decisive interaction condition × time revealed a difference between conditions for the 20 min session only, but not the baseline, where we assumed equal results. Since one could question the direct comparison of the 5 min baseline with the 20 min session, physiological baseline and session parameters were also calculated separately and subjected to *t*-tests. Means, SDs, and within-subject differences between conditions calculated with a dependent-sample *t*-test (two-tailed) for the physiological variables are presented in Table 2 and Table 3. Importantly, those physiological variables, which following a significant interaction of the factors condition x time in the ANOVA show a significant t-test difference between meditation and the story condition, as well as between the meditation condition and the baseline before meditation, are treated as sensitive for an assumed meditation effect.

#### 3.1.1. 2 × 2 Repeated-Measures ANOVA

The 2 × 2 repeated-measures ANOVA for the breathing period (BR) revealed significant main factors of condition (*F*_1,14_ = 4.888, *p* = 0.044) and time (*F*_1,14_ = 13.934, *p* = 0.002), as well as for the decisive interaction condition x time (*F*_1,14_ = 6.803, *p* = 0.021).

Neither the two main effects nor the time-by-condition interactions showed significant differences (all *p* > 0.2) for the heart-rate variability variables RMSSD and HF.

The DFA α-1 levels (sympathetic activity; non-linear fractal parameter) were not different for the main-effect condition (*F*_1,21_ = 2.138, *p* = 0.158), but they differed significantly for the main effect of time (*F*_1,21_ = 7.058, *p* = 0.015) and the interaction condition × time (*F*_1,21_ = 6.803, *p* = 0.021). For the DFA α-2 levels (parasympathetic activity), there were no significant main effects, and the interaction showed no significant difference (all *p* > 0.05).

Regarding the complexity of the heart-rate series, decisive interaction was significant for the variables of approximate entropy (ApEn) and sample entropy (SampEn). The main factor condition did not differ for ApEn (*F*_1,21_ = 1.917, *p* < 0.181), but both the factor time (*F*_1, 21_ = 42.152, *p* < 0.001) and the interaction between the main factors were significantly different (*F*_1, 21_ = 10.177, *p* = 0.004). Regarding SampEn, the main factor condition revealed a significant difference (*F*_1, 21_ = 5.442, *p* = 0.030), but not the factor time (*F*_1, 21_ = 2.030, *p* = 0.169); the interaction condition x time showed a significant difference (*F*_1, 21_ = 15.205, *p* < 0.001).

#### 3.1.2. *T*-Tests between Meditation and Story Conditions

Here, we account for post hoc differences following the ANOVA for differences between meditation and story reading at the (a) 5 min baseline before the experimental sessions and (b) during the 20 min experimental sessions (these differences can partly explain the interaction effect found in the ANOVA; see also Section 3.1.3). Focusing first on the intra-subject baseline comparison with *t*-tests, as expected, no significant differences were found for the 5 min baseline (BL) measure (before the two sessions), but several t-test differences appeared for the variables between the two sessions of meditation vs. reading the story (see below for exact t-test statistics).

Results of the intra-subject session differences during the 20 min sessions (meditation vs. reading a story) showed a significant difference between conditions with respect to the mean duration of the breathing period (BR: breathing period; *t*(15) = 2.87, *p* = 0.012, *d* = 0.91). Participants had longer breathing intervals during the meditation session (M = 4.88; SD = 0.84) than while reading the story (M = 4.18; SD = 0.69).

The RMSSD (*t*(21) = −0.15, *p* = 0.881) reflecting the HRV in the time-domain and HF *t*(21) = −1.12, *p* = 0.281) reflecting the HRV in the frequency domain showed no differences between the two conditions. Note that six individuals had respiratory rates below nine cycles per minute during the meditation condition and therefore had to be excluded from the frequency domain HRV analysis (see methods section). Only 16 individuals were computed in the dependent-sample *t*-test for HF.

However, there were significant differences between conditions with respect to the nonlinear (fractal) parameters of the HRV signal, namely alpha 1 (α 1; *t*(21) = 2.74, *p* = 0.012, *d* = 0.60) and alpha 2 (α 2; *t*(21) = −2.78, *p* = 0.011, *d* = 0.72). While meditating participants showed higher α-1 levels (sympathetic activity; M = 1.35; SD = 0.23) than while reading the story (M = 1.17; SD = 0.35), α-2 levels (parasympathetic activity) were lower during meditation (M = 0.27; SD = 0.10) than while reading the story (M = 0.35; SD = 0.12).

There were significant differences between conditions regarding the complexity of the heart-rate series, namely approximate entropy (ApEn; *t*(21) = −3.27, *p* = 0.004, *d* = 0.42) and sample entropy (SampEn; *t*(15) = 3.50, *p* = 0.002, *d* = 0.71). Participants had lower complexity levels of ApEn and the SampEn during meditation (ApEn: M = 1.22; SD = 0.20; SampEn: M = 1.32; SD = 0.27) than while reading the story (ApEn: M = 1.38; SD = 0.17; SampEn: M = 1.55; SD = 0.29).

#### 3.1.3. T-Tests between Time Intervals

Here, we account for post hoc differences following the ANOVA between the time intervals *t*2 (experimental session) and *t*1 (baseline) for the respective conditions (meditation, story). With one exception, significant *t*-tests for differences in the physiological variables between the baseline (BL) before the experimental session and the respective meditation and story sessions were determined for the meditation session only (these differences can partly explain the interaction effect found in the ANOVA; see also Section 3.1.2.). See Table 3 for details.

There was a significant difference between BL before meditation and the meditation session with respect to the mean duration of the breathing period (BR: breathing period; *t*(17) = 4.277, *p* = 0.001, *d* = 0.88). Participants had longer breathing intervals during the meditation session (M = 4.88; SD = 0.84) than at BL before meditation (M = 4.22; SD = 0.64).

During meditation, participants had higher α-1 levels (sympathetic activity; M = 1.35; SD = 0.23) than during BL before meditation (M = 1.16; SD = 0.26); (*t*(21) = 3.741, *p* = 0.001, *d* = 0.77).

There were significant differences between story reading (M = 1.38; SD = 0.17) and BL before story reading (M = 1.08; SD = 0.17) regarding the first measure of complexity, approximate entropy (ApEn; *t*(21) = 8.405, *p* = 0.001, *d* = 1.76); ApEn was higher during story reading than in the BL before story reading.

There were significant differences between meditation (M = 1.32; SD = 0.27) and BL before meditation (M = 1.47; SD = 0.28) regarding the second measure of complexity, sample entropy (SampEn; *t*(21) = −4.049, *p* = 0.001, *d* = 0.55); SampEn was significantly lower during meditation than during BL before meditation.

To control the validity of the data output above, we created a difference (Δ) score for the two conditions (meditation and control) by subtracting the values of the 5 min baseline (*t*1) from the 20 min session (*t*2). For each of the seven physiological variables, we then calculated a two-sample paired *t*-test between these two Δ scores (Δ meditation, Δ story). Mirroring the results in Table 3, the following *t*- and *p*-values were calculated:

BR: *t* = 2.608, *p* = 0.021; RMSSD: *t* = −0.906, *p* = 0.375; HF: *t* = −1.291, *p* = 0.218; α 1: *t* = 3.285, *p* = 0.004; α 2: *t* = −0.853, *p* = 0.403; ApEn: *t* = −3.190, *p* = 0.004; SampEn: *t* = 8.121, *p* = 0.001.

### 3.2. Temporal Integration of Metronome Beats

A series of repeated-measures ANOVA with two conditions (meditation session vs. reading session) × two (time: pre-session vs. post-session) factors regarding the temporal integration intervals (TI) for each metronome frequency (2, 1.33, 1, 0.5, and 0.33 s. ISI) were performed to test hypothesis 1: mindfulness–meditation states lead to alterations in the duration of the present moment, as assessed with the metronome task, post- vs. pre-session. These analyses revealed neither main effects nor time-by-condition interactions. Across all frequencies, *p*-values corresponding to the main effects time and condition were all larger than 0.05. *p*-values corresponding to the decisive interaction between time and condition were larger than 0.05 across frequencies. Descriptive statistics for these analyses are presented in Table 4. Note that three individuals were not able to produce meaningful groups (did not understand the task) and therefore were excluded from further analysis.

### 3.3. Subjective Time and Body-Self Boundaries

Means, SDs, and within-subject differences between conditions (calculated with dependent sample t-test; two-tailed) for the PBBS and the STSS state scales are presented in Table 5, as well as in Figure 1 and Figure 2. There is a significant difference between conditions (meditation vs. story) with respect to body–self boundaries (BB; *t*(21) = −3.30, *p* = 0.003, *d* = 0.94), the passage of time (POT; *t*(21) = 2.68, *p* = 0.014, *d* = 0.81), and attention-to-time scales (ATT; *t*(21) = −3.32, *p* = 0.003, *d* = 0.80). Participants perceived their body boundaries as less noticeable during meditation (M = 1.81; SD = 1.25) than while reading the story (M = 3.27; SD = 1.80). They felt time passing more quickly (POT: M = 68.76; SD = 22.56) and directed less attention to time (ATT: M = 22.38; SD = 22.15) during meditation than while reading the story (POT: M = 51.14; SD = 20.85; ATT: M = 43.14; SD = 29.04).

### 3.4. Meditation Condition: Correlations between Physiological Variables and Sense of Time and Body

Pearson correlation coefficients between physiological variables and the subjective time and body boundaries for the meditation condition are listed in Table 6. A negative correlation between ApEn and the passage of time was found (*r* = −0.497, *p* = 0.022). The SampEn was negatively correlated with the passage of time (*r* = −0.507, *p* = 0.019). These results indicate that individuals with higher levels of ApEn and SampEn (indicating higher levels of entropy) estimated time to pass more slowly during the meditation session than while reading the story.

Pearson correlation coefficients between the scores of the Meditation Depth Questionnaire (MEDEQ) and subjective time and body boundaries are listed in Table 7. Weak (insignificant) to moderate correlation coefficients (*p* > 0.05) in the hypothesized direction appear for three variables, which indicates that the deeper the meditation, the weaker the body boundaries, the less intensive the body awareness, and the less attention paid to time.

## 4. Discussion

This study explored the effects of meditation in experienced meditators on the awareness of the present moment, subjective time, and the bodily self while also assessing the physiological parameters of heart and breathing rates. We compared a sample of 22 experienced mindfulness meditators before and after a 20 min meditation session (experimental condition) and a 20 min reading session (control condition) in a within-subject design. There were several differences in the physiological variables and subjective reports between the two conditions.

Concerning physiological indices, the breathing rate was reduced during meditation compared to the reading condition and regarding the meditation condition and baseline recording before meditation (i.e., the breathing periods were longer, indicating a more relaxed state often reported during meditation) [17]. Regarding heart-rate variability (HRV), the indices RMSSD and HF did not differ between the two conditions and in comparison to the respective baseline measurements. The nonlinear (fractal) parameters of the HRV signal (detrended fluctuation analysis; DFA) showed higher DFA α-1 during the meditation condition than while reading the story and regarding the baseline before meditation. α-1 relates to sympathetic activity, which was increased during meditation. Participants had relatively lower levels of SampEn during meditation, reflecting decreased heart-rate variability (HRV) complexity, which is related to parasympathetic modulation; SampEn was decreased during meditation as compared to the baseline measure before meditation. In a complementary fashion, participants demonstrated relatively higher levels of ApEn during the story condition than in the meditation condition (reflecting increased HRV complexity) and as compared to baseline measurement before the story condition. How do we interpret these different nonlinear measures of HRV? A time series containing many repetitive patterns (the decreased ApEn) has a relatively smaller complexity (more predictability) [42]. Regarding this interpretation of the data, meditative states with a smaller ApEn thus have a more predictable (less complex) HRV. In comparison, higher entropic, less predictable brain states are typically found in individuals with altered states of consciousness under the influence of psychedelics [43].

Like other induction methods of altered states of consciousness, meditation cannot be conceived in simple, uniform physiological categories [44]. Meditation is not a state of uniform passive relaxation but of concentrated wakefulness influenced by the specifics of the meditation technique. An increase in parasympathetic (slower breathing rate) and sympathetic parameters (higher DFA α-1 levels), which we found in our study, can often be simultaneously recorded [21,22,24]. We showed that a mix of physiological parameters (heart and breathing rates) is significantly different during meditation compared to a control reading session. Correlational statistics with moderately strong, but not significant, correlation coefficients after α correction for multiple testing revealed that individuals with lower HRV complexity (ApEn and SampEn related to parasympathetic modulation) estimated time to pass faster during the meditation session.

Regarding the operationalization of present-moment awareness, the metronome task showed no significant differences between the two conditions. In a similar pre–post-longitudinal study with less experienced meditators in an inter-subject design with a meditation and a control condition (passive listening to a meditation instruction and a recorded story, respectively), meditation led to an increase in the duration of integration intervals at the slowest metronome frequency (inter-stimulus interval, ISI = 3 s) [17]. Although there was a tendency toward an increased integration interval at our slowest metronome frequency (ISI = 2 s), which would have corroborated the earlier findings, this difference was not significant. One possibility for the null finding is that some individuals had difficulties understanding the task and then grouping the metronome beats into meaningful chunks. This methodologic problem occurs in some individuals who are not musically trained, do not have an intuitive grasp of the task, and therefore cannot automatically integrate the beats [45]. Future studies could present the metronome beats in a passive setup without requiring participants to explicitly create auditory groups and to record auditory-evoked potentials in the brain, which reflect the automatic accentuation [46]. Another potential interpretation is that very experienced meditators who are also in a state of relaxation while reading a text easily reach a ceiling effect of present-moment awareness, which would preclude a detectable difference compared to the meditation state.

A different kind of measure for the experienced present utilizing ambiguous figures, such as the Necker cube [47], revealed that experienced meditators could volitionally control the switching of the two perspectives of the Necker cube, i.e., they dwell longer on one perspective before it switches [48,49,50]. No significant effect of the switching rate after vs. before meditation as controlled by a condition of listening to a story was found in one study with beginners and moderately experienced meditators [45]. It remains open whether operationalizations of the present moment using other tasks can detect changes through meditation which are related to present-moment awareness.

In participants’ subjective reports on time and the bodily self, they were less aware that their body boundaries were less salient during meditation than while reading the story; they also felt time passing more quickly, and they directed less attention to time. In complementing subjective reports, systematic qualitative studies have also shown how the awareness of time is diminished in unison with the experienced self in experienced meditators [2,5]. Quantitative studies on time perception during a meditation session have been undertaken with meditation-naïve subjects [6,7]. Our study may be the first to quantitatively assess subjective time and self during the meditation session and investigate these changes among comparatively experienced meditators. These results correspond with earlier findings determined from meditation-naïve individuals instructed to adhere to yogic mindfulness meditation. In comparison to a control condition where the same participants listened to classical music, meditation led to a relative underestimation of duration and a faster subjective passage of time [6].

The major limitation of our study is the number of participants (n = 22). As can be seen in the correlations between different measures, moderate correlation coefficients did not reach the significant level after α adjustment, which is indicative of an underpowered sample size. Since a quantitative study of subjective time and self on such an experienced meditation group has not been conducted so far, we could not resort to empirically available effect sizes. However, empirical data on the subjective variables testing naïve subjects with a different meditation technique showed effect sizes around 0.6. Since, in the present study, we were dealing with extremely experienced meditators, we expected to have larger effect sizes, which we did. Physiological effect sizes for meditation effects reached a Cohen’s *d* of 0.88, and subjective variables on time and self reached a *d* of 0.81. We recruited as many experienced meditators as possible in a reasonable time during several instances of institutional lockdown during the COVID-19 pandemic. The number of participants was at least sufficient to detect the intra-subject differences between conditions for physiological and subjective variables. Researchers who have the possibility to recruit a larger quantity of experienced meditators can use our results as a starting point for further inquiries into the changes of self and time through meditation.

The quantitative measurement of altered states of consciousness provides researchers with a means to investigate the unsolved riddle of consciousness through the investigation of time and self-awareness. Using different induction methods such as psychedelics and other drugs, meditation, or flotation-REST [44,51,52], researchers have started to study altered states of consciousness and the underlying physiological systems dynamics. To sum up our study with exceptionally experienced meditators, our results are in accordance with the knowledge of other psychological or pharmacological induction types of altered states of consciousness where in peak states, subjective time and self are temporarily changed, reduced, or even lost [3,4,53,54,55]. That is, we show the same phenomenon in a quantitative way with the psychological induction technique of meditation, resulting in a joint modulation of the experience of self and of time. Future studies might further assess the larger variety of self-transcendent experiences which we probed with our measures here and assess more precisely different aspects of self [56]. Moreover, future studies could use phenomenologically informed methods to juxtapose (neuro-) physiological data with subjective experiences of time and self in highly experienced meditators [57].

## 5. Conclusions

To our knowledge, this is the first quantitative experimental study with experienced meditators to show how the experience of time and the sense of the bodily self are decreased during self-induced meditation. On the physiological level, the 20 min meditation led to typical paradoxical effects pertaining to signs of increased parasympathetic activity (decreased heart-rate variability complexity and breathing rate) as well as increased sympathetic activity (higher DFA α-1 levels). Accordingly, meditation has both elements of relaxation and concentrated wakefulness. Since we did not find changes in present-moment awareness after versus before meditation as measured with the metronome task, in further endeavors, researchers may further attempt to objectify this important experiential aspect of meditation.

## Figures and Tables

**Figure 1 biology-11-01116-f001:**
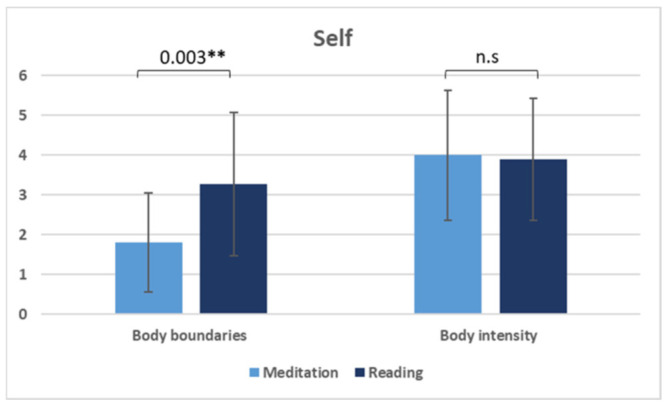
Mean values (error bars reflect SD) for the body boundaries and body intensity scales (y-axis values on a Likert-type scale between 0 and 6). The significant effect (**) for the variable body boundaries and the absent effect for the variable body intensity (not significant, n.s.), according to *t*-tests between the two conditions (meditation and reading), implies less salient body boundaries during meditation.

**Figure 2 biology-11-01116-f002:**
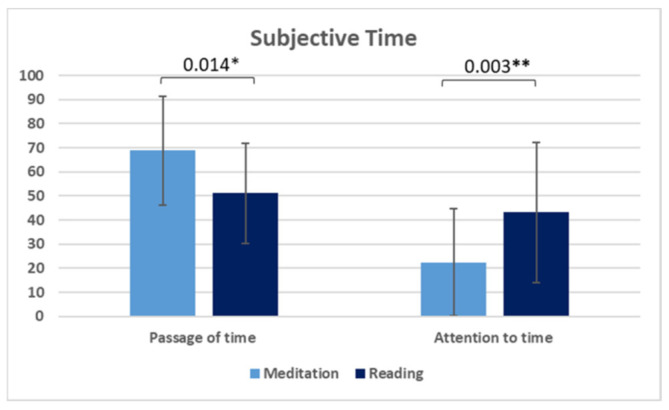
Mean values (error bars reflecting SD) for the passage-of-time and attention-to-time scales (y-axis values on a 100 mm VAS). Significant *t*-tests (* *p* < 0.05, ** *p* < 0.01) between the two conditions (meditation and reading) indicate a faster passage of time and less attention paid to time during meditation.

**Table 1 biology-11-01116-t001:** Descriptive statistics of the study group (n = 22).

Variable	Value
Age (mean ± SD)	47.2 ± 12.45
Gender: female (%)	12 (55)
Educational level:	
high-school degree n (%)	7 (31.8)
university degree n (%)	15 (68.2)
Meditation experience:	
lifetime in years (mean ± SD)	19.2 ± 14.83
lifetime in hours (mean ± SD)	2994.2 ± 3213.44
hours per week (mean ± SD)	4.0 ± 2.08

**Table 2 biology-11-01116-t002:** Descriptive statistics for the physiological variables for the 5 min baseline (BL) measured before the meditation and story condition, respectively, and for the two 20 minute meditation and story sessions. A difference score (Δ) indicates the value differences between the meditation and the story condition. Significant *t*-test coefficients for the differences between the two conditions: * *p* < 0.05, ** *p* < 0.01 are marked **in bold**; ^FDR^ significant after FDR correction for 14 tests; ANOVA signifies a significant interaction effect in the repeated-measures ANOVA.

Physiological Variable 5 Minute Baseline (BL)	Meditation Condition (n = 22)	Story Condition (n = 22)	Δ Score: Meditation-Story	*t =*	*p* =
BL BR: breathing period [s] (mean ± SD)	4.22 ± 0.64	4.23 ± 0.70	−0.01	−0.034	0.974
BL RMSSD (mean ± SD)	29.36 ± 17.74	27.35 ± 13.82	2.01	0.382	0.415
BL HF (mean ± SD)	39.04 ± 19.90	36.94 ± 18.92	2.10	0.406	0.689
BL α 1 (mean ± SD)	1.16 ± 0.26	1.18 ± 0.29	−0.02	−0.380	0.707
BL α 2 (mean ± SD)	0.29 ± 0.14	0.33 ± 0.19	−0.04	−0.915	0.370
BL ApEn (mean ± SD)	1.13 ± 0.24	1.08 ± 0.17	0.05	0.873	0.392
BL SampEn (mean ± SD)	1.47 ± 0.28	1.49 ± 0.33	−0.02	−0.285	0.779
20 min experimental sessions					
Session BR: breathing period [s] (mean ± SD)	4.88 ± 0.84	4.18 ± 0.69	0.70	**2.87**	**0.012 *** ^FDR, ANOVA^
Session RMSSD (mean ± SD)	27.84 ± 11.88	28.23 ± 14.19	−0.39	−0.15	0.881
Session HF (mean ± SD)	33.11 ± 17.49	38.10 ± 24.54	−4.99	−1.12	0.281
Session α 1 (mean ± SD)	1.35 ± 0.23	1.17 ± 0.35	0.18	**2.74**	**0.012 *** ^FDR, ANOVA^
Session α 2 (mean ± SD)	0.27 ± 0.10	0.35 ± 0.12	−0.08	**−2.78**	**0.011*** ^FDR, b^
Session ApEn (mean ± SD)	1.23 ± 0.20	1.38 ± 0.17	−0.15	**−3.27**	**0.004 **** ^FDR, ANOVA^
Session SampEn (mean ± SD)	1.32 ± 0.27	1.55 ± 0.29	−0.23	**3.50**	**0.002 **** ^FDR, ANOVA^

**Table 3 biology-11-01116-t003:** Descriptive statistics for the physiological variables for the 5 min baseline (BL) measure and for the two 20 minute meditation and story sessions. A difference score (Δ) indicates the value differences between the 5 min baseline (BL) and the 20 minute experimental session for the respective meditation and story conditions. Significant *t*-test coefficients for the differences between the two conditions: *** *p* < 0.001 are marked **in bold**; all are significant after FDR correction for 14 tests. ANOVA signifies a significant interaction effect in the repeated-measures ANOVA.

Physiological Variable	Session	5 Min Baseline (BL)	20 Min Sessions	Δ Score: 20 Min. Session–5 Min. BL	*t =*	*p =*
BR: breathing period [s] (mean ± SD)	Meditation	4.22 ± 0.64	4.88 ± 0.84	0.66	**4.277**	**0.001 ***** ^ANOVA^
	Story	4.23 ± 0.70	4.18 ± 0.69	−0.05	−0.763	0.458
RMSSD (mean ± SD)	Meditation	29.36 ± 17.74	27.84 ± 11.88	−1.52	−0.607	0.550
	Story	27.35 ± 13.82	28.23 ± 14.19	0.88	0.625	0.539
HF (mean ± SD)	Meditation	39.04 ± 19.90	33.11 ± 17.49	−5.93	−1.342	0.200
	Story	36.94 ± 18.92	38.10 ± 24.54	1.16	0.646	0.528
α 1 (mean ± SD)	Meditation	1.16 ± 0.26	1.35 ± 0.23	0.19	**3.741**	**0.001 ***** ^ANOVA^
	Story	1.18 ± 0.29	1.19 ± 0.35	0.01	−0.307	0.762
α 2 (mean ± SD)	Meditation	0.29 ± 0.14	0.27 ± 0.10	−0.02	−0.643	0.527
	Story	0.33 ± 0.19	0.35 ± 0.12	0.02	0.711	0.485
ApEn (mean ± SD)	Meditation	1.13 ± 0.24	1.23 ± 0.20	0.10	**2.004**	**0.058**
	Story	1.08 ± 0.17	1.38 ± 0.17	0.30	**8.405**	**0.001 ***** ^ANOVA^
SampEn (mean ± SD)	Meditation	1.47 ± 0.28	1.32 ± 0.27	−0.15	**−4.049**	**0.001 ***** ^ANOVA^
	Story	1.49 ± 0.33	1.55 ± 0.29	0.06	1.381	0.182

**Table 4 biology-11-01116-t004:** Descriptive statistics for temporal integration (TI) at different metronome frequencies (n = 19) for the two conditions (meditation, story).

Variable	Time	Meditation Condition	Story Condition
TI at 2 s. ISI (mean ± SD)	Pre-	3.26 ± 1.66	3.15 ± 1.38
	Post-	3.57 ± 1.95	3.05 ± 1.68
TI at 1.33 s. ISI (mean ± SD)	Pre-	2.87 ± 1.19	2.94 ± 1.37
	Post-	2.48 ± 1.08	2.80 ± 1.16
TI at 1 s. ISI (mean ± SD)	Pre-	2.05 ± 0.62	2.57 ± 1.12
	Post-	2.26 ± 1.19	2.15 ± 0.83
TI at 0.5 s. ISI (mean ± SD)	Pre-	1.63 ± 0.92	1.57 ± 0.93
	Post-	1.44 ± 0.83	1.63 ± 0.99
TI at 0.33 s. ISI (mean ± SD)	Pre-	1.26 ± 0.83	1.26 ± 0.85
	Post-	1.04 ± 0.63	0.88 ± 0.45

**Table 5 biology-11-01116-t005:** Descriptive statistics and independent *t*-test significance between conditions for body and time scales. ^a^ t-Test if not otherwise indicated: * *p* < 0.05, ** *p* < 0.01, marked **in bold**; ^FDR^ significant after FDR correction; ^b^ for Cohen’s *d* see text.

Variable	Meditation Condition	Story Condition	*p*-Values ^a^
Body boundaries (mean ± SD)	1.81 ± 1.25	3.27 ± 1.80	**0.003 **** ^b; FDR^
Body intensity (mean ± SD)	4.00 ± 1.63	3.90 ± 1.54	0.883
Passage of time (mean ± SD)	68.76 ± 22.56	51.14 ± 20.85	**0.014 *** ^b; FDR^
Attention to time (mean ± SD)	22.38 ± 22.15	43.14 ± 29.04	**0.003 **** ^b; FDR^

**Table 6 biology-11-01116-t006:** Correlations between physiological variables, subjective time, and body boundaries during the meditation condition. Significant correlation coefficients marked **in bold**: * *p* < 0.05 (two-tailed); *b* = not significant after α correction.

Variable	BR	α 1	α 2	ApEn	SampEn
Body boundaries	−0.297	−0.041	−0.222	0.095	0.080
Body intensity	0.009	0.148	0.105	0.019	0.018
Attention to time	0.154	0.312	−0.402	0.112	0.042
Passage of time	0.297	0.273	−0.029	**−0.497 * ^b^**	**−0.507 * ^b^**

**Table 7 biology-11-01116-t007:** Correlations (all *p* > 0.05) between meditation depth (MEDEQ) on the one hand and subjective time and body boundaries on the other hand.

Variable	MEDEQ
Body boundaries	−0.384
Body intensity	−0.332
Attention to time	−0.338
Passage of time	0.104

## Data Availability

The data presented in this study are available in the Appendix A.

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
