# Peer review of "Changes in Subjective Time and Self during Meditation"

_biology, 2022, doi:10.3390/biology11081116_

Round 1
Reviewer 1 Report
I understand that subjective experience or responses from the subjects could be a limitation of that research. However, in this article heart rate and breathing rates are also measured which are objective signs. This article is a comprehensive explanation of physiological effects of meditation. I have no further recommendations.
Author Response
Rev: I understand that subjective experience or responses from the subjects could be a limitation of that research. However, in this article heart rate and breathing rates are also measured which are objective signs. This article is a comprehensive explanation of physiological effects of meditation. I have no further recommendations.
Answer: We appreciate the positive evaluation of our work.
Reviewer 2 Report
This paper details the possible effects of meditation on self-awareness and time perception by designing two sets of experiments. It has studied the influence of meditation on people's psychology and consciousness in the form of quantitative study, which is a very meaningful study.
1.The title of the article may need to be revised for better understanding by readers.
2. In Table 1, the lifetime hours of formal meditation training’s mean value is 2994.2 while the SD is ±3213.44, the absolute value of the standard deviation is greater than the mean. In addition, the error bars in Figure 1 and Figure 2 are also relatively large. Please explain the reasons.
3. Line 183 Please be specific about how each participant is kept alone, as they may interact with each other if they get too close. Please confirm whether each person enters a separate room for the test, or each participant maintains a certain distance in a large room?
4. In Section 3.1, it describes many experimental results, but there is only one table, and many data only appear in the text, which is not convenient for readers to read and understand.
5. Line 379 The paragraph numbering is incorrect.
6. The units of the ordinates in Figures 1 and 2 are not marked, and there are no specific values marked in the bar chart. They may need to be improved so that readers can easily understand.
Author Response
This paper details the possible effects of meditation on self-awareness and time perception by designing two sets of experiments. It has studied the influence of meditation on people's psychology and consciousness in the form of quantitative study, which is a very meaningful study.
Answer: We appreciate the positive evaluation of our study.
Rev: 1.The title of the article may need to be revised for better understanding by readers.
Answer: We have changed the title to: “Changes in subjective time and self during meditation”
Rev: 2. In Table 1, the lifetime hours of formal meditation training’s mean value is 2994.2 while the SD is ±3213.44, the absolute value of the standard deviation is greater than the mean. In addition, the error bars in Figure 1 and Figure 2 are also relatively large. Please explain the reasons.
Answer: The sample is skewed to the right as we have a few exceptionally experienced meditators. The three most experienced participants had 11.040, 9.105, and 8.724 hours of experience. As a comparison, the median lies at 1755.5 hours but these three individuals increased the standard deviation.
Rev: 3. Line 183 Please be specific about how each participant is kept alone, as they may interact with each other if they get too close. Please confirm whether each person enters a separate room for the test, or each participant maintains a certain distance in a large room?
Answer: We have specified the condition more precisely in that each participant was alone in the room and meditating by him or herself. Under 2.3 General procedure we added: “Each participant was tested individually.” That way no interaction was happening between participants who came on separate days to the lab.
Rev: 4. In Section 3.1, it describes many experimental results, but there is only one table, and many data only appear in the text, which is not convenient for readers to read and understand.
Answer: Table 2 was created to give an overview of the statistics (ANOVA, t tests). We have therefore added a sentence in the paragraph prior to the statistics and wrote: “The means and standard deviations for the four cells referring to the 2 x 2 ANOVA for each variable (BL, experimental session; meditation, reading story) can be read in Table 2.” We changed Table 2 in that we more clearly mark the difference between the baseline and the experimental conditions by inserting an extra table line. In addition we added a new Table 3 to accommodate the complex results.
Rev: 5. Line 379 The paragraph numbering is incorrect.
Answer: We fixed the incorrect subtitle numbering 3.3. and 3.4.
Rev: 6. The units of the ordinates in Figures 1 and 2 are not marked, and there are no specific values marked in the bar chart. They may need to be improved so that readers can easily understand.
Answer: We have added in the Figure captions for the two Figures the information: “y-axis values on a Likert-type scale between 0 and 6” (Fig. 1); “y-axis values on a 100 mm VAS“ (Fig. 2).
Reviewer 3 Report
I am reviewing “Changes in Subjective Time and Self but not the Present Moment after Meditation” for Biology. I like the concept of the paper, the literature is fine, and the data are good. However, the analyses need to change; I will outline the changes. Furthermore, the tables are confusing. In addition, I wonder why the authors used experienced meditations rather than novices. I will also make many suggestions for grammar.
The authors want to correlate some of the measures, but they need to correlate all of them or none of them. I argue that the authors should correlate every measure at time 1 and time 2 should be correlated to every measure at time 1 and time 2. One could make an argument that the measures that are conceptually related and significantly correlated to each other should be examined using MANOVA, which addresses alpha level inflation caused by multiple comparisons. The authors correctly say that few correlations were found due to the small sample size in the discussion.
The second change to the analyses is very important. The 2 x 2 repeated-measures analysis simply does not work because the analyses that the authors offer are not focused; they compare measures across meditation conditions at time 1 and time 2 and then compare time 1 to time 2 for each measure within conditions. The authors make all these comparisons because some of the analyses did not show a difference at time 1 and other analyses showed a difference at time 1. The lack of a difference at time 1 across meditation conditions allows a clean comparison of the measure across meditation conditions, but the presence of a difference at time 1 across meditation conditions does not allow for that clean comparison, so the authors need to pivot and compare the measure across time within a condition. The analyses can be much more focused with a simple solution. Specifically, the authors should calculate the difference score from time 1 to time 2 (subtract time 2 from time 1) and then they can compare those scores across conditions in a paired-samples t-test. As I mentioned previously, the measures that are conceptually related (and significantly related) should be analyzed in a MANOVA first; if the authors’ statistics program does not run a repeated-measures MANOVA, they can calculate z-scores for all the difference score measures and then average those z-scores and analyze them across the two meditation groups. If the MANOVA is significant, the individual t-tests can be run, and the authors controlled for family-wise alpha error.
The tables are very confusing. The authors should put in the difference scores for each meditation condition along with the t and p values. The authors could also put in scores at time 1 and time 2 but comparisons do not need to be made across time or conditions. The comparison of the difference scores provides the main point, which is if the meditation condition produced a greater change in a measure than the reading condition.
Why did the authors evaluate experienced meditators? I assume the reason pertains to the idea that experienced meditators would show larger effects than novices, but I do not remember the authors providing the reason that they used experienced meditators rather than novices. The authors should provide the reason for testing this group at the end of the introduction and they should talk about replicating the current study with experts and novices and provide the expected results in the future research portion of the discussion.
I will tackle the grammar suggestions in order from the beginning to the end of the paper. Line 15 should put commas around “therefore”. Line 21 should end the sentence that ends on that line “in the meditation condition”. The next sentence should start “Meditation participants”. Line 23 should say “This study is the first one to show that the sense of self”.
Line 59 should say “investigated the fact that time perception”. Line 62 should say “This finding indicated”. Line 65 should say “10 minutes” and line 66 should say “1 second”. Line 67 should say “pre-meditation compared to”. Line 68 should say “for 10 minutes”. Line 71 should say “We were interested in the way experienced meditators would relate their impression of time to their sense of self. Line 73 should say “diminished, in that the meditation session overall passes”. Line 75 should say “slow passage of”. The sentence starting on line 77 should be deleted.
Line 94 should say Multiple path analyses found that”. The sentence on line 97 should not say “the higher the heart-rate variability, the longer the duration…” because a comma is acting as a conjunction (perfectly natural in speech though). Instead, the sentence could say “high heart-rate variability was related to long duration….” The information is at the bottom of the page, so I cannot see duration of…whatever.
Line 110 should put commas around “therefore”. Line 118 should put a parenthesis after “conditions” and autonomic activity (several calculated…the time series), yield alterations…task (widening of heard metronome intervals), and changes/reductions subjective time and perceived salience of body boundaries.” Line 140 should say “criteria included engaging in regular meditation practice for at least 5 years with at least two meditation sessions occurring”. The footnote on page 3 should say “Because this study was the first one”.
Line 146 should put parentheses around the mean and SD information. Line 147 should say “mouth; they were fluent…reported no history”. Line 157 should say “one questionnaire about…variables and another one about meditation practice.” Line 163 should say “Participants were”. Line 166 should say “session, participants”. Line 176 should say “condition), participants”. Line 180 should say “using their own tempo without moving.”
Line 186 should say “Finland), which is a three-channel…recorder that also records”. Line 188 should say “The ECC data were acquired …electrodes were positioned…configuration: one was placed…and the other one was placed”. Line 191 should say “data were”. Line 199 should say “Likert-type scale”. Line 209 should say “A high score”. Line 210 should say “low score signifies a weak distinction”. Line 220 should say “A high score represents a high”.
Line 233 should say “Participants were also”. Line 237 should say “is, thus, defined…onsets (with a presented…is assumed)”. Lines 239-241 should say “showed that a sustained brain response was initiated at each of the accents when participants…metronome beat.” I have no idea how “the group onset” fits into that sentence. Line 243 should say “time 1”. Line 246 should say “, with each one presented”. Line 247 should say “tendency measures”.
Line 260 should say “The ECG signals”. Line 262 should say “Data were first”. Line 271 should say “following paragraph.” Line 279 should say “might not represent and convey the same biological”. Line 280 should say “We, therefore, proceeded”. Line 281 should say “The RMSSD…in heart rate.” Line 283 should say “and it is considered to be free”. Line 287 should say “RMSSD, it is thought”. Line 293 should put a comma after “parameters”. Line 298 should say “time series and it is referred to as consecutive”.
If the authors choose to keep their results in their current form, I would (in addition to refusing to review the paper) need to make many comments about the results in expressing that the results demonstrated a significant main effect of time or a significant main effect of session type or a significant Time x Session Type interaction. If the authors run their analyses and set up the tables the way I suggested, the tables will provide all the important information and then they can simply provide a combined Results and Discussion section where the authors start by talking about the measures showing significant differences across session type: meditation and reading. I admittedly skimmed the discussion because the results need to be radically changed. In my perusal, the discussion seemed sufficient. The authors should talk about using experienced meditators rather than novices and they should talk about future research replicating the current study with both experts and novices and they should provide the expected results.
I enjoyed reading this paper and I think that the authors could provide a very compelling, clear, and simple story with the recommended changes.
Author Response
Rev: I am reviewing “Changes in Subjective Time and Self but not the Present Moment after Meditation” for Biology. I like the concept of the paper, the literature is fine, and the data are good. However, the analyses need to change; I will outline the changes. Furthermore, the tables are confusing. In addition, I wonder why the authors used experienced meditations rather than novices. I will also make many suggestions for grammar.
Answer: Thank you for the positive evaluation of our paper. We respond to the suggestion of a different data analysis below, we have changed the Table slightly and wrote an explanatory sentence in the text (also according to the suggestion of reviewer 2). We chose experienced meditators because we expected stronger effects. Loosing body boundaries and the sense of time is reported by experienced meditators. Novices might report a stronger sense of time and self as they are yet not accustomed to the meditation techniques as they are beginners. We wanted to find out what experienced meditators feel and report.
Rev: The authors want to correlate some of the measures, but they need to correlate all of them or none of them. I argue that the authors should correlate every measure at time 1 and time 2 should be correlated to every measure at time 1 and time 2. One could make an argument that the measures that are conceptually related and significantly correlated to each other should be examined using MANOVA, which addresses alpha level inflation caused by multiple comparisons. The authors correctly say that few correlations were found due to the small sample size in the discussion.
Answer: The suggestion of this reviewer resembles an exploratory attempt to see what can be found in the data regarding 60 variables, or to reduce it to the variables in Table 2, 28 variables. Our aim was to concentrate on the two time variables and two self variables (the content of our study design next to the metronome task) and conduct confirmatory analysis (also with respect to avoid inflation of the error rate (alpha). We, therefore, correlated the four variables of concern (2 time, 2 self variables) with those 5 physiological variables which had shown to be sensitive for differences between conditions (meditation, story) (see Table 2 for significant effects, Table 3 for these correlations). Since we had a considerable variance in meditation experience, we correlated the four subjective variables also with meditation depth (now Table 7).
Regarding the question of MANOVA, with this method we can examine any number of dependent variables simultaneously with one test. Compared to several individual ANOVAs (where we have to adjust the alpha level), we increase the statistical power of the tests. However, after consulting statistics manuals, the more dependent variables we include in a MANOVA, the larger the sample size must be in order to be able to make reliable statements. With our n = 22 subjects, a series of ANOVA seems to be the more appropriate way. Since the statistical power was strong enough, we found 5 out of 7 physiological variables to be significant after alpha adjustment.
Rev: The second change to the analyses is very important. The 2 x 2 repeated-measures analysis simply does not work because the analyses that the authors offer are not focused; they compare measures across meditation conditions at time 1 and time 2 and then compare time 1 to time 2 for each measure within conditions. The authors make all these comparisons because some of the analyses did not show a difference at time 1 and other analyses showed a difference at time 1. The lack of a difference at time 1 across meditation conditions allows a clean comparison of the measure across meditation conditions, but the presence of a difference at time 1 across meditation conditions does not allow for that clean comparison, so the authors need to pivot and compare the measure across time within a condition. The analyses can be much more focused with a simple solution. Specifically, the authors should calculate the difference score from time 1 to time 2 (subtract time 2 from time 1) and then they can compare those scores across conditions in a paired-samples t-test. As I mentioned previously, the measures that are conceptually related (and significantly related) should be analyzed in a MANOVA first; if the authors’ statistics program does not run a repeated-measures MANOVA, they can calculate z-scores for all the difference score measures and then average those z-scores and analyze them across the two meditation groups. If the MANOVA is significant, the individual t-tests can be run, and the authors controlled for family-wise alpha error.
Answer: This is how we see the case and thus opted to use a 2 x 2 repeated-measures ANOVA. With this statistical method we control for potential differences between conditions (factor condition: meditation vs. story) as well as for differences between t2 and t1 (factor time). The interaction between the two factors shows a difference across time for only one of the conditions (the decisive information). Through post-hoc t tests and the descriptive values (Table 2) we can infer the direction of the difference. That is the standard way of controlling for effects. The reviewer writes: “the presence of a difference at time 1 across meditation conditions does not allow for that clean comparison, so the authors need to pivot and compare the measure across time within a condition.” The reviewer suggests to employ difference scores between t2 and t1 for each variable. But that that is what our baseline calculations between conditions at t1 show, in that before both sessions (meditation, story reading) the physiological variables did not differ (Table 2). Therefore, we should be able to use the ANOVA in a meaningful way.
However, we completely agree with the reviewer that a thorough presentation of post-hoc effects is necessary. As inspired by this reviewer, we, therefore, follow a complementary approach. We leave the ANOVA as the starting point of our calculations and then proceed in calculation post-hoc t tests for the factor condition (meditation vs. story) and for the factor time interval (baseline vs. experimental session). That way, we have all information, including the one the reviewer requested. Accordingly, we have created a difference score, we show the t values, and we have created a new Table to account for the differences in the post-hoc t tests. We hope to satisfy this reviewer as all difference scores are openly visible. This has led to the expansion of the results section and a novel Table.
Rev: The tables are very confusing. The authors should put in the difference scores for each meditation condition along with the t and p values. The authors could also put in scores at time 1 and time 2 but comparisons do not need to be made across time or conditions. The comparison of the difference scores provides the main point, which is if the meditation condition produced a greater change in a measure than the reading condition.
Answer: We now added the differences scores for meditation minus story. This should be the main difference of interest, the difference between conditions. This is a valid comparison because the two conditions do not differ significantly at t1 (baseline). We provide the descriptive values between t2 and t1 for each condition as well as the difference scores discussed by the reviewer as important. In addition we provide the t test coefficients in this table. Moreover, we have created a new Table 3 which contains the other difference scores (as well as t statistic coefficients) for the physiological variables for the 5-minute baseline (BL) measure and for the two 20-minutes’ meditation and story sessions, respectively. This analysis complements the ANOVA and leads to a thorough analysis of all effects between the factors of experimental conditions (meditation, story) and time (session vs. baseline).
Rev: Why did the authors evaluate experienced meditators? I assume the reason pertains to the idea that experienced meditators would show larger effects than novices, but I do not remember the authors providing the reason that they used experienced meditators rather than novices. The authors should provide the reason for testing this group at the end of the introduction and they should talk about replicating the current study with experts and novices and provide the expected results in the future research portion of the discussion.
Answer: We inserted the following two sentences into the last paragraph of the introduction: “We chose more experienced meditators because we expected to find stronger effects for the sense of self and time as well as for the subjectively experienced present moment. Losing the sense of body boundaries and the sense of time is often reported by very experienced meditators.”
Rev: I will tackle the grammar suggestions in order from the beginning to the end of the paper. Line 15 should put commas around “therefore”. [corrected] Line 21 should end the sentence that ends on that line “in the meditation condition”. [corrected] The next sentence should start “Meditation participants”. [corrected] Line 23 should say “This study is the first one to show that the sense of self”.
Line 59 should say “investigated the fact that time perception”. Line 62 should say “This finding indicated”. [corrected] Line 65 should say “10 minutes” [corrected] and line 66 should say “1 second”. [corrected] Line 67 should say “pre-meditation compared to”. Line 68 should say “for 10 minutes”. Line 71 should say “We were interested in the way experienced meditators would relate their impression of time to their sense of self. [corrected] Line 73 should say “diminished, in that the meditation session overall passes”. [corrected] Line 75 should say “slow passage of”. [corrected] The sentence starting on line 77 should be deleted. [corrected]
Line 94 should say Multiple path analyses found that”. [corrected] The sentence on line 97 should not say “the higher the heart-rate variability, the longer the duration…” because a comma is acting as a conjunction (perfectly natural in speech though). Instead, the sentence could say “high heart-rate variability was related to long duration….” [corrected] The information is at the bottom of the page, so I cannot see duration of…whatever. [corrected]
Line 110 should put commas around “therefore”. [corrected] Line 118 should put a parenthesis after “conditions” and autonomic activity (several calculated…the time series), yield alterations…task (widening of heard metronome intervals), and changes/reductions subjective time and perceived salience of body boundaries.” [all corrected] Line 140 should say “criteria included engaging in regular meditation practice for at least 5 years with at least two meditation sessions occurring”. [corrected] The footnote on page 3 should say “Because this study was the first one”. [corrected]
Line 146 should put parentheses around the mean and SD information. Line 147 should say “mouth; they were fluent…reported no history”. [corrected] Line 157 should say “one questionnaire about…variables and another one about meditation practice.” [corrected] Line 163 should say “Participants were”. Line 166 should say “session, participants”. [corrected] Line 176 should say “condition), participants”. Line 180 should say “using their own tempo without moving.” [corrected]
Line 186 should say “Finland), which is a three-channel…recorder that also records”. [corrected] Line 188 should say “The ECC data were acquired …electrodes were positioned…configuration: one was placed…and the other one was placed”. [corrected] Line 191 should say “data were”. Line 199 should say “Likert-type scale”. [corrected] Line 209 should say “A high score”. [corrected] Line 210 should say “low score signifies a weak distinction”. [corrected] Line 220 should say “A high score represents a high”. [corrected]
Line 233 should say “Participants were also”. [corrected] Line 237 should say “is, thus, defined…onsets (with a presented…is assumed)”. [corrected] Lines 239-241 should say “showed that a sustained brain response was initiated at each of the accents when participants…metronome beat.” [corrected] I have no idea how “the group onset” fits into that sentence. [corrected] Line 243 should say “time 1”. Line 246 should say “, with each one presented”. Line 247 should say “tendency measures”. [corrected]
Line 260 should say “The ECG signals”. [corrected] Line 262 should say “Data were first”. Line 271 should say “following paragraph.” [corrected] Line 279 should say “might not represent and convey the same biological”. [corrected] Line 280 should say “We, therefore, proceeded”. [corrected] Line 281 should say “The RMSSD…in heart rate.” [corrected] Line 283 should say “and it is considered to be free”. [corrected] Line 287 should say “RMSSD, it is thought”. [corrected] Line 293 should put a comma after “parameters”. Line 298 should say “time series and it is referred to as consecutive”. [corrected]
Answer: In the majority of cases we have corrected the phrases as indicated by this reviewer and marked the case with a [corrected] sign. We thank the reviewer for this thorough work.
Rev: If the authors choose to keep their results in their current form, I would (in addition to refusing to review the paper) need to make many comments about the results in expressing that the results demonstrated a significant main effect of time or a significant main effect of session type or a significant Time x Session Type interaction. If the authors run their analyses and set up the tables the way I suggested, the tables will provide all the important information and then they can simply provide a combined Results and Discussion section where the authors start by talking about the measures showing significant differences across session type: meditation and reading. I admittedly skimmed the discussion because the results need to be radically changed. In my perusal, the discussion seemed sufficient. The authors should talk about using experienced meditators rather than novices and they should talk about future research replicating the current study with both experts and novices and they should provide the expected results.
Answer: Regarding the comment about results being highlighted as “significant main effect of time or a significant main effect of session type or a significant Time x Session Type interaction”, our Results section on page 8 are presented in that way. We have changed the Discussion section according to the reviewer’s suggestion and it now incorporates the results from the Tables important for a thorough analysis.
Rev: I enjoyed reading this paper and I think that the authors could provide a very compelling, clear, and simple story with the recommended changes.
Answer:
We appreciate the positive evaluation.
Round 2
Reviewer 3 Report
I am reviewing for a second time “Changes in Subjective Time and Self but not the Present Moment after Meditation” for Biology. I made some major suggestions in my first review, and I asked 1) the authors to make all those changes or 2) the editors to not ask me to review again. The authors chose to make some major changes and they told the editor that they made the necessary changes, but they did not. I will try to make my suggestions very clear, and I will provide the rationale for those suggestions. If the authors do not make all the changes, I will recommend rejection, so the editor should either reject the paper or find a new reviewer. I hope I am being very clear. I make a few comments for the introduction and discussion, and I make very few grammar suggestions, because I focused on the analytic changes that were and were not made and the latter reduced my motivation to add clarity to the prose.
I am going to start by describing the way I review for the editorial board and the authors because something is getting lost in translation. I want to help authors publish and I most often recommend revise and resubmit and publication (I have rejected 4-6 papers out of the 300-400 papers I have reviewed) when the researchers set forth on a plan and they carry it out in a paper, and they follow important recommendations. I focus on measurement, design, analyses, and grammar and I try to help authors improve their paper. If I am sure about something, I state it. If I am not, I state that the authors should check on the issue. I write in a very formal way (no demonstratives or contractions or ending sentences with prepositions or using a comparative like greater and not providing the comparison, etc.) and I give authors suggestions to write formally as well, but the authors and editor can make decisions about the grammar that they will and will not use. However, if authors bring in a design confound (the authors do not bring in design confounds in the current paper) and I find it and they do not change it, I recommend rejection. If the editors publish that paper, I do not review for that journal again. If authors use the wrong design or statistics, I tell them about it and I recommend rejection if they do not listen. I figure that I am asked to help improve the paper as an expert, and I should either be removed as a reviewer, or the authors should follow the directions, or they should provide excellent reasons for not following the suggestions. As I said in the first review, even though this paper uses an incredibly small sample size, I believe it will show significant differences that answer the research question if the authors use the proper statistics and present them in a simple, efficient, clear, and straightforward way.
So, we need to start with the research question. The research question is “Does meditation affect the various dependent variables compared to a control condition, which is reading?” Many other studies examine this same research question. For example, many meditation studies examine the effect of meditation on anxiety. However, these studies usually use a 2 (Meditation Condition) x Time (Pre-Meditation and Post-Meditation) design, with meditation condition as a between factor and time as a repeated-measures factor. The good studies covary out pretest scores from the posttest scores and then they compare the posttest scores across meditation conditions with one-way ANCOVAs, and they answer the research question simply. However, most studies examine and describe main effects of meditation and time as well as the interaction of these two factors. The problem with this analysis is that main effects of meditation or time do not provide an answer to the research question. In fact, a significant interaction of meditation and time alone does not answer the research question either, as a break down of the simple main effects of the interaction is needed to answer the research question. Specifically, the simple main effects should show no significant difference across meditation conditions (meditation and control) at time 1, but they should show a significant difference across meditation conditions at time 2 or they should show no significant change across time for the control condition but a significant change across time for the meditation condition. The problem with the latter finding is that the two meditation conditions are not compared. Nevertheless, this finding generally shows that meditation produced an effect that the control condition did not. A worse finding is that both meditation and control conditions show significant changes across time. With such a finding, researchers must attempt to argue that the effect of the meditation condition across time is larger than the effect of the control condition across time. However, researchers provide no statistical comparison for the argument; they simply argue that the t-value for the meditation comparison is larger than the t-value for the control condition, but is a t-value of 4 significantly larger than a t-value of 3? A statistical comparison (evidence) is needed. Instead, the authors should have used the covariate analysis to provide a simple and clear answer to their research question. I have reviewed similar types of studies, and I recommended using covariate analyses or rejection if the authors did not listen. Researchers should use the right tool for the job and simple analyses, answers, and interpretations are better than complex and convoluted ones.
The current study uses a completely repeated-measures design, which brings in two possible simple analyses. First, the current study could treat the meditation condition as a between-factor and covary out the pretest scores from the posttest scores and compare the mediation condition to the control condition for the posttest scores using ANCOVAs. The problem is that the authors would be treating a repeated-measures factor as a between-groups factor, which does not reflect reality. Second, the authors can calculate the difference scores across time (subtract score at time 2 from score at time 1) for the control group (reading) and the meditation group and then compare those difference scores. If the difference in the difference scores across meditation conditions is significant, the answer to the research question is that meditation does produce a different effect than reading.
The authors did not follow this simple and effective suggestion for evaluating and presenting results with the goal of answering the research question in a simple, clear, and efficient way. Rather, the authors ran a 2 x 2 repeated-measures ANOVA followed by one or both pairs of simple main effects comparisons followed by some difference-score comparison that is presented in a confusing way; I cannot understand why or how the authors are subtracting the reading condition from the meditation condition. In fact, I suggested that the authors should present all the measures at time 1 and time 2 along with their difference scores for time (time 1 – time 2) across the reading and meditation conditions as well as a paired-samples t-test of those difference scores across reading and meditation conditions. Instead, I have no idea what Table 2 presents because it says it is subtracting the scores for reading from the scores for meditation. Are the authors making that comparison across time 1 or time 2 or are they doing a double subtraction score across both times? I know that the presentation does not make sense. Again, the difference score for time 1 and time 2 should be calculated for the meditation condition and the reading condition and then those difference scores should be presented in a table and then a comparison of those difference scores should be presented with the t-value.
On a few different notes, the authors say that they do not want to compare all the measures in correlations due to their small sample and the potential for alpha errors, but they are perfectly fine comparing all their measures in a series of t-tests without controlling for alpha error, which can be done using MANOVA. The authors should pick a lane and stay in it (be consistent in their arguments). If correlations are presented, they should be presented first and those measures (at least those measures, perhaps all measures) should be compared using MANOVA. Alternatively, the authors should present all the correlations without worrying about alpha error (as most researchers do, although some researchers run their alpha level at .01) and then they can compare all the difference scores (from time 1 to time 1) across meditation and control conditions using many t-tests (or they could run their alpha level at .01).
On page 2, the authors should briefly provide the reason that they were interested in experienced meditators (i.e., more knowledge). As a final point before grammar suggestions, the authors did not use the correct language to describe main effects and interactions. As an example, lines 338-339 should say “main effects of condition….as well as a significant Condition x Time interaction”.
As for a few grammar suggestions, lines 14 and 15 should say “2 sessions”. Line 20 should say “In the meditation condition, participants subjectively”. Line 23 should say “more quickly compared to the control condition.” Line 23 should say “This study is the first one to show how” to avoid using a demonstrative “this” in “This is”. This what? Lines 32-33 should say “A mixed pattern of increased…was found during meditation” to avoid starting a sentence with “There is” as sentences in English should start with a noun, which is a person, place, or thing, and “there” is a place, but it is not used as such in this type of phrase. Line 37 should say “This experiment is the first one” to avoid using a demonstrative.
Lines 84 and 90 should avoid using “We” to keep the authors’ identities anonymous. Lines 95-96 say “higher, longer, higher, and greater” but higher than what? The authors should provide the comparison or say “high, long, high, and great”. Lines 109-110 should put the “i.e.,” information in parentheses. Line 113 could say “participants” instead of “they”. Line 126 should say “used” instead of “involved”. Line 128 should say “2 consecutive days”. Line 138 should put a comma after “Initially”. Line 142 is confusing, but I think the authors should remove “as” to clarify the prose.
As I said in the first review, I am happy to read the paper again if the authors make the changes. If they do not, I will recommend rejection. Therefore, the editors should read the paper and the authors replies before sending it out to me to review if the editors want to publish the paper.
Author Response
Reviewer: Specifically, the simple main effects should show no significant difference across meditation conditions (meditation and control) at time 1,
Answer: The main effect of condition is a result of between-group differences at time 2 (not time 1): The significant interaction effect “time x condition” was followed up by post hoc tests, which showed no differences between conditions for time 1 (BL), but several significant differences for time 2 (Session). If the reviewer means that there should not be a difference between meditation and the control condition at time 1, this is exactly the finding which can be seen in Table 2 in the upper part (5-minute baseline (BL)) referring to the statistics in the text.
Reviewer: but they should show a significant difference across meditation conditions at time 2 or they should show no significant change across time for the control condition but a significant change across time for the meditation condition.
Answer: In Table 2 and Table 3 (which we added according to the new statistics demanded by the reviewer) we show both of these results: (1) the difference between meditation and the control condition for 5 out of 7 variables (Table 2), and (2) the detailed changes across time for the meditation condition for 3 variables and for 1 variable for the control condition. Following from these insightful analyses which follow the criticism of the reviewer we now can identify 3 physiological variables which are changed in the meditation condition but not the control condition; these are discussed in the Discussion section.
Although not explicitly stated by the reviewer or us authors in these discussions with the reviewer, we apply this reasoning only to the physiological variables because for the subjective variables we have only one time point, after the 20-minute interventions.
Reviewer: The problem with the latter finding is that the two meditation conditions are not compared.
Answer: We are puzzled by this sentence since we have only one meditation condition vs. a control condition. We assume that the reviewer refers to these two conditions as the meditation conditions (in plural).
Reviewer: Nevertheless, this finding generally shows that meditation produced an effect that the control condition did not. A worse finding is that both meditation and control conditions show significant changes across time.
Answer: In one physiological variable (ApEn) we find a change over time in the control condition. That is not “good” or “bad”; this is an empirical finding for one variable which we discuss. It merely shows that the reading condition is not a passive intervention but has an effect over time. Against such an effect we investigate the effects of meditation.
Reviewer: With such a finding, researchers must attempt to argue that the effect of the meditation condition across time is larger than the effect of the control condition across time. However, researchers provide no statistical comparison for the argument; they simply argue that the t-value for the meditation comparison is larger than the t-value for the control condition, but is a t-value of 4 significantly larger than a t-value of 3? A statistical comparison (evidence) is needed.
Answer: The combination of the interaction effect in the 2 × 2 ANOVA (alone it does not answer the question) plus the two t tests in both directions (across condition at t1 and t2) plus for the two conditions separately across t1 and t2 gives us the vital information. Nevertheless, we have now exactly done what the reviewer suggested: “researchers must attempt to argue that the effect of the meditation condition across time is larger than the effect of the control condition across time”. See below for these t tests for delta values for meditation (t2-t1) – control condition (t2-t1) for the physiological variables. Not surprisingly, we find very similar results as before, since the delta values build upon the former analysis.
Reviewer: I suggested that the authors should present all the measures at time 1 and time 2 along with their difference scores for time (time 1 – time 2) across the reading and meditation conditions
Answer: This already had been done in the statistics which are summarized in Table 2 and 3.
Reviewer: as well as a paired-samples t-test of those difference scores across reading and meditation conditions.
Answer: This has now been additionally done, see below.
Reviewer: Instead, I have no idea what Table 2 presents because it says it is subtracting the scores for reading from the scores for meditation.
Answer: The subtraction leads to a difference score, which is exactly the point estimate on which the post hoc paired t-tests reported in Table 2 are based. Please note that this is a within-subject design, so difference scores between conditions make sense.
Reviewer: Are the authors making that comparison across time 1 or time 2 or are they doing a double subtraction score across both times? I know that the presentation does not make sense. Again, the difference score for time 1 and time 2 should be calculated for the meditation condition and the reading condition and then those difference scores should be presented in a table and then a comparison of those difference scores should be presented with the t-value.
Answer: This procedure is an add-on to the standard way of presenting the data as we did, following ANOVAs and looking at the decisive interaction effect and additionally to present the t test for the conditions and time points separately, as done and highlighted in the Tables 2 and 3. We used the classic way of treating the data and profited from the first round of criticism as we added Table 3. With Table 2 and Table 3 we have all effects visible (i.e. effects for meditation that are not found in the control condition). Still, the t tests between the difference scores are now available in the text (see below).
Using the difference values as suggested by the reviewer should necessarily lead us to very similar results as the one already presented. We created difference scores for meditation t2-t1 and control conditions t2-t1 (MedDelta and StoryDelta and then calculated a t test between these two variables). These new variables build on the former variables and should not show drastically different results. We have done this for the physiological variables and the results are as expected (mirroring the results in Table 3):
T tests (paired samples) for delta values for meditation (t2-t1) – control condition (t2-t1) for the physiological variables:
BR: breathing period: t = 2.608, p = 0.021
RMSSD: t = -0.906, p = 0.375
HF: t = -1.291, p = 0.218
α 1: t = 3.285, p = 0.004
α 2: t = -0.853, p = 0.403
ApEn: t = -3.190, p = 0.004
SampEn: t = 8.121, p = 0.001
We now add this information just below Table 3.
Reviewer: On a few different notes, the authors say that they do not want to compare all the measures in correlations due to their small sample and the potential for alpha errors, but they are perfectly fine comparing all their measures in a series of t-tests without controlling for alpha error, which can be done using MANOVA. The authors should pick a lane and stay in it (be consistent in their arguments). If correlations are presented, they should be presented first and those measures (at least those measures, perhaps all measures) should be compared using MANOVA. Alternatively, the authors should present all the correlations without worrying about alpha error (as most researchers do, although some researchers run their alpha level at .01) and then they can compare all the difference scores (from time 1 to time 1) across meditation and control conditions using many t-tests (or they could run their alpha level at .01).
Answer: We have already answered in the previous reply why we did not opt for MANOVA, for reasons of statistical assumptions, which can be read in statistics text books. This procedure would be an odd way of presenting the correlations for subjective variables which are only filled out by participants at t2 (and not at t1 as this reviewer claims). The use of simple correlations is the most straightforward for this kind of investigation and data (as witnessed by numerous other studies in the field).
Reviewer: On page 2, the authors should briefly provide the reason that they were interested in experienced meditators (i.e., more knowledge).
Answer: This is the paragraph, were we describe our rationale (we added “experienced” to meditators):
We were interested in the way experienced meditators would relate their impression of time to their sense of self during the meditation session. Experienced meditators typically report that the sense of time together with the sense of self is diminished, in that the meditation session overall passes subjectively very quickly. During the first minutes of meditation induction, the focused attention on the bodily self might lead to a slow passage of time, but the whole session is typically judged to have passed relatively quickly for experienced meditators, as the sense of self is diminished.
Reviewer: As for a few grammar suggestions, lines 14 and 15 should say “2 sessions”. Line 20 should say “In the meditation condition, participants subjectively”. Line 23 should say “more quickly compared to the control condition.” Line 23 should say “This study is the first one to show how” to avoid using a demonstrative “this” in “This is”. This what? Lines 32-33 should say “A mixed pattern of increased…was found during meditation” to avoid starting a sentence with “There is” as sentences in English should start with a noun, which is a person, place, or thing, and “there” is a place, but it is not used as such in this type of phrase. Line 37 should say “This experiment is the first one” to avoid using a demonstrative.
Lines 84 and 90 should avoid using “We” to keep the authors’ identities anonymous. Lines 95-96 say “higher, longer, higher, and greater” but higher than what? The authors should provide the comparison or say “high, long, high, and great”. Lines 109-110 should put the “i.e.,” information in parentheses. Line 113 could say “participants” instead of “they”. Line 126 should say “used” instead of “involved”. Line 128 should say “2 consecutive days”. Line 138 should put a comma after “Initially”. Line 142 is confusing, but I think the authors should remove “as” to clarify the prose.
Answer: We have integrated several of these suggestions.
Round 3
Reviewer 3 Report
The authors did not listen to my comments, so I recommend rejection as I said in my previous review.